# Analysis of the Behavior of Low-Noise Asphalt Mixtures with Modified Binders under Sinusoidal Loading

**DOI:** 10.3390/ma15165476

**Published:** 2022-08-09

**Authors:** Roman Pacholak, Andrzej Plewa, Wladyslaw Gardziejczyk

**Affiliations:** Faculty of Civil Engineering and Environmental Sciences, Bialystok University of Technology, 15-351 Bialystok, Poland

**Keywords:** low-noise pavement, stiffness modules, rubber granulate, modified bitumen, crumb rubber

## Abstract

The paper presents the results of tests of the stiffness modulus according to the 4PB-PR method of low-noise asphalt mixtures with the addition of rubber granulate (RG). Mixtures of this type are characterized by an increased air void content (about 10–25%). This causes a rapid bitumen oxidation, which results in oxidative hardening, contributing to a faster deterioration of the properties of the mixtures. This means that binders of appropriate quality should be used in the process of producing asphalt mixtures, which will provide the mixtures with sufficiently high technical properties. The tested asphalt mixtures are differentiated according to the type of bitumen modifiers: styrene–butadiene–styrene copolymer (SBS) and crumb rubber (CR). The article presents the tests results of the stiffness modulus using the 4PB-PR method. This test has a high correlation with regard to “in situ” tests. The research proved that each of the modifiers used increased the stiffness modulus of low-noise asphalt mixtures. Replacing the mineral aggregate with 30% RG leads to a tenfold decrease in the stiffness modulus. In the entire range of analyzed temperatures, mixtures with the use of modifiers show higher values of the elastic component of the stiffness modulus, as evidenced by lower values of the phase angle.

## 1. Introduction

The 20th century was characterized by the rapid development of road traffic [1]. During this period, road pavements built in Poland, due to the type of structure and the materials used, did not fully reduce the noise that was increasing all the time. This phenomenon attracted the attention of many researchers and contributed to the creation of low-noise asphalt pavements. Such pavements include, among others: BBTM (French for Beton Bitumineux Tres Mince), SDA (Semi-Dense Asphalt), PA (Porous Asphalt), noise reduced stone mastic asphalt SMA LA (German for Lärm Arm), PERS (Poroelastic Road Surface) [2]. Research has shown that the use of such mixtures for the wearing course of road pavements allows the reduction of noise to 10–12 dB [3,4]. Asphalt mixtures from this group are characterized by air void content in the range of 15–25%. Many studies [5,6] have shown that higher air void content contributes to the formation of carbonyl groups in the binder, resulting in oxidative hardening, which causes faster pavement degradation. Therefore, to produce “quiet” mixtures, it is required to properly select bitumen binders with high technical parameters, particularly with the increased resistance to long-term (in-service) aging [7].

The quality of the raw material used in the distillation process of the crude oil has a huge impact on the properties of road bitumen, which was used in the production of asphalt mixture. Unfortunately, in recent years there has been a trend to refine light crude oil (increasing fuel profit), which has reduced the distillation residue from 20% to 10% or less. For this reason, the quality of industrially produced bitumen does not meet the requirements for use in “quiet” pavements, which has been confirmed by investigations carried out by many scientific centers [8,9,10]. Therefore, in order to improve the performance properties of neat binders, various types of modifiers are often used. This group includes: chemical modifiers, polymers, hydrocarbons, nanomaterials, fillers, and oxidants. The best-known types of the modifiers from these groups are: styrene–butadiene–styrene copolymer (SBS), crumb rubber (CR) derived from scrap tires, or the more and more often used composite of these two-SBS/CR modifiers [11,12,13].

CR obtained from scrap tires was first used in the 1960s by engineer Charles McDonald. It is a well-known additive to bitumen binders. The process of bitumen modification with CR is a complex process and depends on many factors, such as mixing time and temperature, number of rotations during mixing, method of grinding rubber and its particle size, method of adding, chemical composition of binder, and rubber [14,15]. In the modification process at the temperature of 180 °C, the rubber partially undergoes devulcanization and swelling, which are caused by the absorption of light aromatic structures contained in the binder [16]. According to the analyses described in [17,18,19,20], rubber-asphalt binder improves properties of asphalt mixtures, such as rutting resistance, freezing-thawing resistance, aging resistance, low temperature cracking, and fatigue resistance.

SBS copolymer is the most commonly used modifier for bitumen binders. It began to be used on a large scale as bitumen modifier in the 1970s of the 20th century [21]. SBS copolymer, when added to bitumen, changes its chemical properties: an interaction begins between the bitumen and the polystyrene and polybutadiene blocks. Interactions between polybutadiene blocks are carried out by π-electrons of positively charged groups of bitumen, and they are much stronger than interactions between polystyrene blocks, which interact with electron-rich groups through aromatic protons [22]. The effect of this interaction is hardening of bitumen binder, thanks to which it is possible to lower the breaking point, decrease the rate of penetration, extend the viscoelastic range, improve fatigue life and rutting resistance, reduce the temperature sensitivity, and increase the stiffness of asphalt mixtures [23,24,25,26].

Recently, composites have been increasingly used in place of single bitumen additive. One of them is the polymer-rubber-asphalt binder SBS/CR. Due to the simultaneous modification, such a composite is able to eliminate the disadvantages and strengthen the advantages characteristic of a single modifier, e.g., only SBS or only CR [27]. In the case of the SBS/CR composite, the storage stability of binder in the high temperature range, its aging resistance and low temperature cracking resistance were improved [28,29].

It is necessary in the design process to properly select the technical parameters of asphalt mixtures in order to construct durable, safe, and characterized-by-low-maintenance-costs road pavements [30]. One of the most important characteristics of asphalt mixtures is the stiffness modulus. It mainly determines the structural durability of road pavements. It allows one to predict the “behavior” of asphalt mixtures in road surface construction under the traffic, particularly at the place of tire/pavement contact [31,32]. The stiffness modulus of asphalt mixtures is a key parameter used in the design process of the fatigue life of road pavements (low temperature cracking and rutting resistance) in terms of mechanical and empirical analyses. Low values of the stiffness modulus at low test frequencies testify to poor pavement resistance to permanent deformation, while its high values at higher frequencies negatively affect the low-temperature performance at winter conditions [33]. The higher values of the stiffness modulus make it possible to reduce the tensile stresses that arise at the bottom of the asphalt layer (the criterion of cracking of asphalt layers) and to reduce the compressive stresses on the upper asphalt layer [34]. It is known that the tensile stresses at the bottom of the layer are almost twice as high as the compressive stresses at the top of the layer [35]. There are several methods for determining the value of the stiffness modulus of asphalt mixture, e.g., indirect tensile modulus test (IDT), uniaxial compressive test (UC) or four point bending with prismatic samples (4PB-PR). The research on the stiffness modulus of asphalt mixtures proved that the 4PB-PR method allows one to simulate the loading conditions of mixtures similar to the real pavement working conditions [36].

However, the influence of rubber granulate (RG) on changes in stiffness modulus by the 4PB-PR method of “quiet” mixtures has not yet been investigated, which is an important aspect in the design of this type of pavement.

The aim of the research presented in this paper is to analyze the effect of the type of modification of bitumen binders with SBS polymer and CR and the addition of RG (1/4 mm fraction) on the change of technical parameters (stiffness modulus and phase angle) in the four-point bending study with prismatic samples 4PB-PR of low-noise asphalt mixtures.

Chapter 2 presents the characteristics and properties of the materials used for the designed asphalt mixtures (PA8, SMA8 and SMA8 LA) and mineral-rubber-asphalt mixtures SMA8 LA (10% RG), SMA8 LA (20% RG), SMA8 LA (30% RG). The grain size distribution (percentage of material passing through the sieve), binder content, air void content, and density of mixtures are given. The basics and principles of determining the stiffness modulus and phase angle according to the four-point bending with prismatic samples (4PB-PR) are also discussed.

Chapter 3 presents and discusses the results of tests of stiffness modulus 4PB-PR for conventional asphalt mixtures and results of tests of stiffness modulus for asphalt mixtures with the addition of RG, change of the phase angle Φ at variable temperature and frequency for conventional asphalt mixtures and change of the phase angle Φ at variable temperature and frequency for asphalt mixtures with the addition of RG. A discussion of the results in comparison with the findings of other researchers presented in the literature is also provided.

## 2. Materials and Methods

### 2.1. Bitumen Binders

Four types of bitumen binders were used for the tests:
−Bitumen 50/70 (reference);−Bitumen 50/70 modified with copolymer SBS (5%) (SBSM-5);−Bitumen 50/70 modified with CR (10%) (CRM-10);−Bitumen 50/70 modified with a combination of copolymer SBS (2%) and CR (10%) (SBSM-2 + CRM-10).

The technical properties of the modified binders before and after the RTFOT (rolling thin film oven test) technological aging process are presented in Table 1. Detailed test results of bitumen binders and aggregates used for asphalt mixtures are described in publications [37,38].

### 2.2. Used Additives for Binders and Asphalt Mixtures

Modified binders were obtained in the “wet process”. Kraton D1192 which is a linear triblock polymer with 30% styrene content by mass, was used for the preparation of polymer bitumen. Information on the basic chemical properties of the SBS copolymer is presented in Table 2. The rubber-asphalt binder was prepared on the basis of CR with a grain size of 0/0.8 mm. The grain size analysis of the CR is presented in Table 3.

In the mixtures with increased flexibility, part of the aggregate in the “dry process” was replaced with RG with a grain size of 1/4 mm. The sieve analysis of the granulate is presented in Table 4.

Bitumen with a penetration of 50/70 was used for the production of modified binders. Bitumen was heated to a temperature of 180 °C. A measured amount of modifier (5% SBS copolymer or 10% CR, or 2% SBS copolymer and 10% CR) was gradually added to the bitumen and mixed for 1 h at a constant speed of 700 rpm [39]. Then, using the RTFOT device, the binder was subjected to the test simulating the process of technological aging according to [40].

Before being added to the mix, RG was preheated, which caused a swelling effect and a “slight” devulcanization of the rubber to allow the binder to bond with the rubber more effectively. In mineral-rubber-asphalt mixtures, part of the mineral aggregate was replaced with RG in the amount of 10%, 20%, and 30% in relation to the mixture volume in order to improve their flexibility.

### 2.3. Designed Low-Noise Asphalt Mixtures

The following asphalt mixtures and low-noise mineral-rubber-asphalt mixtures were used for laboratory tests: porous asphalt (PA8), stone mastic asphalt (SMA8), stone mastic asphalt reducing tire/road noise (SMA8 LA), and stone mastic asphalt reducing tire/road noise with 10% (SMA8 LA [10% RG]), 20% (SMA8 LA [20% RG]), and 30% (SMA8 LA [30% RG]). The mixtures were prepared in a laboratory mixer which set up a 3-dimensional counter-rotating mixing process with an eccentric rotary agitator. Before compacting, the mixtures were spread into a thin layer and for 2 h subjected to thermostating in a dryer at the temperature of 145 ± 5 °C with the reference binder and at 155 ± 5 °C for the modified binder. Such a procedure made it possible to bring the mixture preparation process closer to the conditions in an asphalt plant and is required by the standard [41]. The samples for 4PB-PR tests were prepared according to [42] in a slab roller compactor with the dimensions of the plates 300 mm × 400 mm × 80 mm. Then, rectangular beams with dimensions of 50 mm × 63 mm × 400 mm were cut from these plates.

The binder content, air void content, grain size composition, and density of the designed asphalt mixtures are presented in Table 5 and Figure 1.

Particle size distribution of tested mixtures (Figure 1) indicate that the tested mixtures are characterized by discontinuous grain size. The presented curves show that the PA8 mixture contains the most air voids, and the SMA8 mixture has the least air voids. When comparing SMA8 LA mixtures, it was concluded that increasing the addition of rubber granulate approximates the grain size curves of SMA8 LA mixtures with granulate to the SMA8 mixture. This is an advantageous solution due to the noisiness of the pavement as the air void content in such mixtures increases compared to SMA8 LA.

### 2.4. Stiffness Modulus and Phase Angle According to the Four Point Bending with Prismatic Samples (4PB-PR)

The stiffness of asphalt mixtures has a significant impact on the stress level generated in the structural layers of the road surface by traffic loads and temperature changes during operation. These stresses determine the resistance to fatigue and cracking of the pavement. The stiffness modules by the four-point bending method were tested according to [43]. The loads applied to the sample were cyclic and sinusoidal, which is typical in real conditions. The thermostatic time at the test temperature was 1 h. According to [43], the samples were rotated 90° along the longitudinal compaction axis in the slab. The stiffness modulus was determined at the 100th load cycle. Due to the fact that asphalt mixture is a typical visco-elastic material, its mechanical properties largely depend on the test temperature and frequency. Therefore, it was decided to perform tests in a wide temperature range: 5 °C, 15 °C, 25 °C and at the following frequencies: 0.5 Hz, 1.0 Hz, 5.0 Hz, 10 Hz, 20 Hz.

The value of the stiffness modulus is defined as the absolute value of the complex modulus, |E *|, determined from the dependency between the stress and the deformation at time t. Characteristic for viscoelastic materials is the shift of the stress curve in relation to the deformation curve by the value of the phase angle, Φ (Figure 2).

Perfectly elastic bodies are characterized by a phase angle, Φ = 0°. In viscous bodies, the phase angle fluctuates around 90°. For visco-elastic bodies, the angle value ranges from 0° to 90°.

The stiffness modulus consists of the real (*E*_1_) and the imaginary (*E*_2_) component:(1)$$\left|E^{*}\right|=\sqrt{E_{1}^{2}+E_{2}^{2}},$$

The real component is determined according to the Equation (2):(2)$$E_{1}=\gamma \cdot \left(\frac{F}{z}\cdot \cos(\Phi )+10^{-6}\cdot \mu \cdot \omega^{2}\right),$$
where:
*γ*—shape coefficient (1/m);*F*—vertical force (kN);*z*—specimen deflection (m);*Φ*—phase angle (deg);*μ*—mass coefficient (kg);*ω*—angular frequency (rad/s).

The imaginary component is calculated according to the Dependency (3):(3)$$E_{2}=\gamma \cdot \frac{F}{z}\cdot \sin(\Phi ),$$

The mass and shape coefficients are calculated according to the Formulas (4)–(7):(4)$$\gamma =\frac{L^{2}}{B\cdot H^{2}}\cdot \left(0.75-\frac{A^{2}}{L^{2}}\right),$$
(5)$$\mu =R\{x\}\cdot \left(\frac{M}{\pi^{4}}+\frac{m}{R(A)}\right),$$
(6)$$R\left\{x\right\}=\frac{12L}{A}\cdot \left(\frac{1}{3\frac{x}{L}-3\frac{x^{2}}{L^{2}}-\frac{A^{2}}{L^{2}}}\right),$$
(7)$$A=\frac{L-l}{A},$$
where:
*L*—distance between outer supports (m);*B*—sample width (m);*H*—sample height (m);*M*—sample mass (kg);*m*—mass of the moving parts of the device (kg);*l*—distance between places of sample loading (m);*A*—distance between the outer clamp and the next inner clamp (m);*x*—position of the probe for measuring beam deformation (m).

Phase angle is defined as the phase difference between stress and strain and is determined by Equation (8):(8)$$\Phi =\left(B_{\varepsilon }-B_{t}\right)\cdot \frac{180}{\pi },$$
where:
*B_ε_*—phase angle of the approximate function of the strain value (rad);*B_t_*—phase angle of the approximate function of the stress value (rad).

## 3. Results and Discussion

### 3.1. Stiffness Modulus 4PB-PR

The results of testing the stiffness modulus with the 4PB-PR method at different frequencies at temperatures 5 °C, 15 °C, and 25 °C are shown in Table 6 and Table 7, while the change of the stiffness modulus values as a function of temperature at 10 Hz is shown in Figure 3.

The results of the 4PB-PR stiffness modulus tests presented in this study confirm the previous research results described in publications [44,45,46,47,48]. The SBS copolymer consists of three-dimensional styrene-butadiene-styrene chains, in which the rigid styrene domains are dispersed in a flexible butadiene matrix that serves as a continuous phase. When SBS is added to bitumen, the styrene blocks, through physical connections, can form three-dimensional networks that enable elastomers to give binders higher stiffness. When crumb rubber is added to the binder, it causes physical swelling and chemical degradation (devulcanization and depolymerization). Maltenes (light fractions of bitumen) are absorbed by rubber networks. This process is the main phenomenon that causes the swelling of crumb rubber (three–five times its original volume). In addition, a “gel” structure is formed at the bitumen/rubber interface. At this stage, effective stiffening of the binder takes place, which consequently affects the value of stiffness modulus of asphalt mixtures.

Due to the fact that asphalt mixtures are materials with visco-elastic properties, their technical features should be presented as a function of temperature change. The stiffness modulus of asphalt mixtures significantly decreases with increasing test temperature. This phenomenon results in a reduction in load-bearing capacity and load-carrying capability. Inverse changes are observed when the test temperature is decreased: the value of stiffness modulus begins to increase rapidly. This indicates that there is an increase in thermal stresses in the visco-elastic material. When the thermal stress value exceeds the tensile strength of the material, the initiation of low-temperature cracking occurs.

When analyzing the results presented in Table 6 and Table 7, it should be clearly stated that the stiffness modulus depends on many factors: type of mixture, type of binder used, amount of RG, test temperature, and frequency.

Considering the type of mixture, the highest values of the stiffness modulus were obtained at 5 °C at the frequency of 20 Hz for the mixture SMA8 with SBSM-2 + CRM-10 and CRM-10 binder (15,423 MPa and 14,978 MPa, respectively). On the other hand, the lowest values of modulus under the same test conditions were obtained for the mixture SMA8 LA (30% RG) with bitumen CRM-10 and SBSM-2 + CRM-10 (951 MPa and 964 MPa, respectively). It is a tenfold decrease in the value of the modulus in relation to the SMA8 LA mixture without the addition of RG (CRM-10-9680 MPa, SBSM-2 + CRM-10–10,057 MPa). It is worth emphasizing that the addition of another 10% of RG to the SMA8 LA mixture causes a decrease in the modulus value by approximately 50% in practically the entire range of analyzed temperatures.

The lowest (favorable) temperature sensitivity (e.g., at the frequency of 10 Hz) in the group of mixtures without the addition of RG achieved the mixtures with SBSM-5 and SBSM-2 + CRM-10 binder (Figure 3). On the other hand, the highest (unfavorable) temperature sensitivity was shown by the mixtures with the reference bitumen 50/70. In mixtures where part of the aggregate was replaced with RG, the lowest sensitivity to temperature changes was found in the mixtures SMA8 LA (10% RG) with bitumen SBSM-5, SMA8 LA (20% RG) with SBSM-2 + CRM-10 binder, and SMA8 LA (30% RG) with CRM-10 binder. However, the most sensitive to temperature are the mixtures of SMA8 LA (10% RG and 20% RG) with bitumen 50/70 and SMA8 LA (30% RG) with the SBSM-5 binder.

Testing the stiffness modulus at variable frequencies of load application in the range from 0.5 Hz to 20 Hz allows for predicting the behavior of asphalt mixtures at variable vehicle speeds. The conducted tests proved that the temperature and the amount of RG have a significant influence on the change of the range of modulus values obtained at different load frequencies. The lowest discrepancies were obtained at 5 °C for mixtures without RG, the highest—at 25 °C for mixtures with 30% RG. This proves that at 5 °C the phase angle is much lower (higher stiffness of asphalt mixtures) than at 25 °C (Table 7). Low values of the stiffness modulus of mixtures with the addition of rubber granulate allow us to conclude that they will be able to transfer and withstand higher tensile stresses, thus allowing us to decrease the low-temperature cracking.

The second degree polynomial was used to describe the changes in the stiffness modulus values:(9)$$\begin{array}{ccc} Z & = & a_{0}+a_{1}X_{1}+a_{2}X_{1}^{2}+a_{3}X_{2}+a_{4}X_{2}^{2}+a_{5}X_{3}+a_{6}X_{3}^{2}+a_{7}X_{4}+a_{8}X_{4}^{2}\\ & + & a_{9}X_{1}X_{2}+a_{10}X_{1}X_{3}+a_{11}X_{1}X_{4}+a_{12}X_{2}X_{3}+a_{13}X_{2}X_{4}\\ & + & a_{14}X_{3}X_{4} \end{array}$$
where:
Z—analyzed mixture parameter (stiffness modulus 4PB-PR);$a_{0}-a_{14}$—regression coefficients;$X_{1}$—type of mixture;$X_{2}$—type of binder;$X_{3}$—temperature (°C);$X_{4}$—frequency (Hz).

The statistical analysis of the obtained results was started with the significance test using the ANOVA analysis of variance (STATISTICA software). The results of this analysis are presented in Table 8.

Based on the analysis of the parameters, it can be clearly stated that the temperature, type of mixture and binder, and frequency are important factors affecting the stiffness modulus, because the *p*-value is lower than the assumed significance level α = 0.05 (*p*-Value < 0.05). Analyzing the square term referring to the type of binder (Type of Binder (Q)) and the factor describing the interaction of the binder type and frequency (2L·4L), no significant influence was found on the values of the stiffness modulus determinations. The values describing the parameters of the regression model are summarized in Table 9.

On the basis of the analysis, it was observed that the value of the corrected determination coefficient was R^2^adj = 76%, which proves the correct adoption of the model.

The developed model of changes in the stiffness modulus can be presented using the following Relationship (10):4PB-PR = −361,542 − 16,414·TM + 135·TM^2^ + 26,015·TB − 51·TB^2^−6973 Temp + 4·Temp^2^ + 3822·F − 7·F^2^ − 142·TM·TB + 68·TM·Temp − 34·TM·F − 6·TB·Temp + 1·TB·F − 3·Temp·F(10)
where:
TM—type of mixture: SMA8 = 105, PA8 = 106, SMA8 LA = 107, SMA8 LA (10% RG) = 108, SMA8 LA (20% RG) = 109, SMA8 LA (30% RG) = 110;TB—type of binder: 50/70 = 101, SBSM-5 = 102, CRM-10 = 103, SBSM-2 + CRM-10 = 104;Temp—test temperature;F—frequency.

The obtained regression model (Equation (10)) allows us to predict with good precision (about 76%) the stiffness modulus of the analyzed mixtures with SBS and crumb rubber modified binder (Figure 3).

The graphical interpretation of the 4PB-PR stiffness modulus change as a function of the type of mixture and binder is shown in Figure 4a, as a function of the binder type and temperature 4(b), and as a function of the binder type and frequency 4(c).

### 3.2. Phase Angle Φ

The changes in the phase angle Φ for the individual mixtures at different temperatures and frequencies are presented in Table 10 and Table 11. A graphical interpretation of the change in phase angle, as a function of frequency at temperature 15 °C is shown in Figure 5.

In the literature, there are few studies of the stiffness modulus tested using the 4PB-PR method. In addition, the authors of these publications hardly analyze the very important parameter, which is the phase angle, Φ. According to [49,50], the phase angle, Φ, mainly defines the viscoelastic properties of asphalt mixtures. It reflects the proportion of viscous and elastic parts in the asphalt mixtures. A lower value of the phase angle, Φ, indicates that elastic properties predominate in the mixture. In the case of mixtures with the base bitumen, the phase angle in the low-frequency area remained almost constant, and then gradually decreased with increasing frequency. A similar relationship was also observed in asphalt mixtures with modified binders, but the values of the phase angle, Φ, are lower in relation to mixtures with base bitumen, which is confirmed by the test results (Figure 5). This proves that asphalt mixtures with modified binders are more rigid (the elastic phase dominates), while the mixtures with a conventional binder are more flexible (lower content of the elastic phase).

When analyzing the obtained results of the phase angle, Φ, on the basis of the stiffness modulus tests using the 4PB-PR method presented in Table 10 and Table 11 and in Figure 5; it should be stated that the type of binder, temperature, and frequency show a significant influence on the change of the phase angle, Φ. Mixtures with modified binders show higher values of the elastic component of the stiffness modulus (lower phase angle, Φ) in the entire range of tested temperatures and frequencies of loading. This fact proves that these mixtures are more resistant to accumulation and deformation in pavements compared to mixtures with conventional binders. For example, when analyzing the values of the phase angle, Φ, in the mixtures without the addition of RG at a temperature of 15 °C (Figure 5), it is clearly visible that the best elastic properties were obtained by all the mixtures with modified binder SBSM-2 + CRM-10 (the smallest value of the phase angle, Φ). No such obvious difference can be observed in mixtures with the addition of RG. This proves that in these mixtures, the value of the phase angle, Φ, is mainly determined by the “elasticity” of RG.

The type of mixture does not significantly affect the value of the phase angle, Φ, in the low frequency range of 0.5 Hz, 1 Hz, 5 Hz and temperatures of 5 °C and 15 °C (Table 9). The greatest changes were observed only for the SMA8 LA (30% RG). On the other hand, the type of modified binder and the amount of RG additives have a significant impact.

Changes in this parameter at higher frequency ranges (10 Hz and 20 Hz) and test temperatures (25 °C) are significantly influenced by the type of binder, the amount of RG addition and the type of mixture.

The comprehensive description of the dependence of the phase angle, Φ, of asphalt mixtures with modified binder was made using the second degree polynomial:
(11)$$\begin{array}{ccc} Z & = & a_{0}+a_{1}X_{1}+a_{2}X_{1}^{2}+a_{3}X_{2}+a_{4}X_{2}^{2}+a_{5}X_{3}+a_{6}X_{3}^{2}+a_{7}X_{4}+a_{8}X_{4}^{2}\\ & + & a_{9}X_{1}X_{2}+a_{10}X_{1}X_{3}+a_{11}X_{1}X_{4}+a_{12}X_{2}X_{3}+a_{13}X_{2}X_{4}\\ & + & a_{14}X_{3}X_{4} \end{array}$$
where:
Z—analyzed mixture parameter (phase angle, Φ);$a_{0}-a_{14}$—regression coefficients;$X_{1}$—type of mixture;$X_{2}$—type of binder;$X_{3}$—temperature (°C);$X_{4}$—frequency (Hz).

The first stage of model evaluation was to perform the significance test using ANOVA. The results of this analysis are presented in Table 12.

The comprehensive statistical analysis of the parameters listed in Table 12 allows us to unequivocally state that the type of mixture and binder, temperature, and frequency are important factors affecting the phase angle, Φ, because the *p*-value related to them is lower than the assumed significance level α = 0.05. This relationship is not only observed with the square expression describing the influence of temperature. The existence of interactions between the type of mixture and the temperature, which affects the analyzed parameter (*p*-value less than α = 0.05), is of significant importance.

The developed regression model of the dependence of the phase angle, Φ, in terms of the type of mixture and binder, temperature, and frequency are summarized in Table 13.

Based on the analysis, it can be concluded that the value of the corrected determination coefficient equals R^2^ adj = 92%, which proves the correct adoption of the model.

The developed model of the analyzed change in the phase angle, Φ, can be presented using the Equation (12):Φ = 13,119.572 − 47.956·TM + 0.134·TM^2^ − 206.074·TB + 0.894·TB^2^ + 4.300 Temp + 0.005·Temp^2^ − 0.536·F + 0.030·F^2^ + 0.202·TM·TB − 0.012·TM·Temp − 0.007·TM·F − 0.020·TB·Temp + 0.003·TB·F − 0.006·Temp·F(12)
where:
TM—type of mixture: SMA8 = 105, PA8 = 106, SMA8 LA = 107, SMA8 LA (10% RG) = 108, SMA8 LA (20% RG) = 109, SMA8 LA (30% RG) = 110;TB—type of binder: 50/70 = 101, SBSM-5 = 102, CRM-10 = 103, SBSM-2 + CRM-10 = 104;Temp—test temperature;F—frequency.

The obtained regression model (Equation (12)) allows to predict with very good precision (about 92%) the stiffness modulus of the analyzed mixtures with SBS and crumb rubber modified binder (Figure 5).

The graphic interpretation of the change of the phase angle as a function of the type of mixture and binder is shown in Figure 6a, as a function of the type of binder and temperature in Figure 6b, and as a function of the type of binder and frequency in Figure 6c.

## 4. Conclusions

Based on the analysis of technical parameters obtained in the four-point bending test with prismatic samples of stone mastic mixtures SMA8, stone mastic asphalt reducing tire/road noise SMA8 LA and porous asphalt PA8 with SBS copolymer-modified binders, crumb rubber modified binders, and binders simultaneously modified with SBS and crumb rubber, the following conclusions were made:The type of modifier has a significant impact on the stiffness modulus of all analyzed asphalt mixtures, increasing its values in the entire range of analyzed temperatures from 5 °C to 25 °C. The highest values of the stiffness modulus were obtained at 5 °C and at the frequency of 20 Hz for the mixture of SMA8 with the binder SBSM-2 + CRM-10 and CRM-10 (15,423 MPa and 14,978 MPa). On the other hand, the lowest values of modulus under the same test conditions were obtained by the mixture SMA8 LA (30% RG), with bitumen CRM-10 and SBSM-2 + CRM-10 (951 MPa and 964 MPa, respectively).The addition of rubber granulate significantly lowers the stiffness modulus. It was found that the addition of 30% rubber granulate reduces the value of the stiffness modulus tenfold. For example: SMA8 LA mixture with bitumen CRM-10 and SBSM-2 + CRM-10 (stiffness modulus-9680 MPa and 10,057 MPa, respectively), SMA8 LA (30% RG) (stiffness modulus-951 MPa and 964 MPa).Replacing the mineral aggregate by another 10% rubber granulate in the SMA8 LA mixture causes a decrease in the stiffness modulus by about 50% in the entire range of analyzed temperatures.Mixtures SMA8, PA8, SMA8 LA with the reference bitumen 50/70 had the highest (unfavorable) temperature sensitivity. The highest temperature sensitivity in mixtures, in which part of the aggregate was replaced with rubber granulate, was achieved by the mixtures SMA8 LA (10% RG and 20% RG) with asphalt 50/70 and SMA8 LA (30% RG) with the SBSM-5 binder. The SMA8 LA (10% RG) mixture with bitumen SBSM-5, SMA8 LA (20% RG) mixture with SBSM-2 + CRM-10 binder and SMA8 LA (30% RG) with CRM-10 binder showed the lowest sensitivity to temperature changes.The type of binder, the temperature, and the frequency show a significant influence on the change of the phase angle, Φ.Mixtures with the use of modified binders show higher values of the elastic component of the stiffness modulus (lower phase angle, Φ) in the entire range of tested temperatures and frequency of loading. This fact proves the greater resistance of these mixtures to accumulation and deformation in a pavement compared to mixtures with conventional binders.According to the analysis of the obtained test results, the optimal solution from a practical point of view is to use a binder simultaneously modified with 2% of SBS and 10% of crumb rubber for the production of low-noise asphalt mixtures. Mixtures SMA8 LA with the addition of 10% of rubber granulate are a good solution for applications in a climate where the average annual temperature is between 5–10 °C.

## Figures and Tables

**Figure 1 materials-15-05476-f001:**
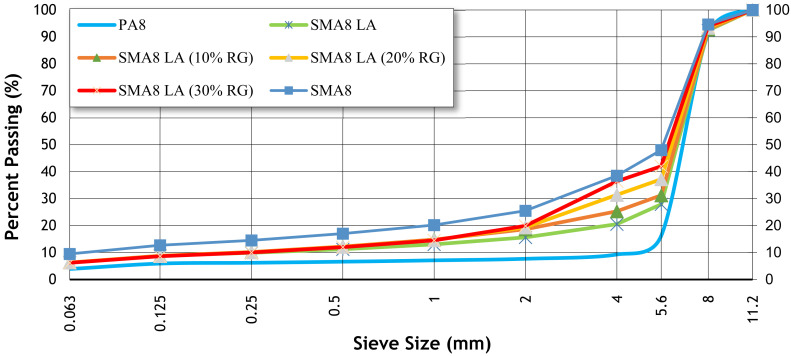
The particle size distribution of tested mixtures [39].

**Figure 2 materials-15-05476-f002:**
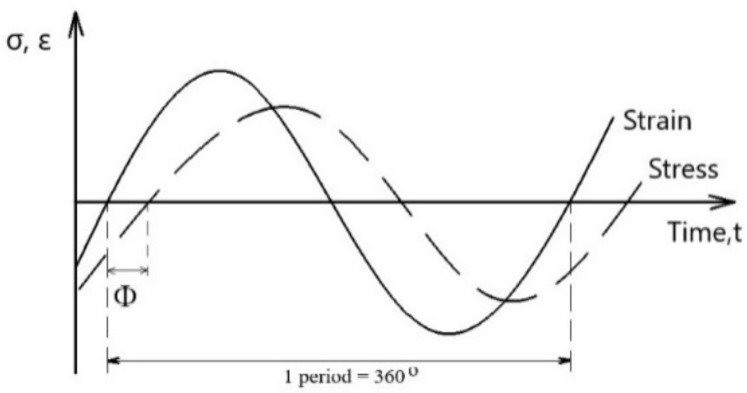
Stress and strain as a function of time.

**Figure 3 materials-15-05476-f003:**
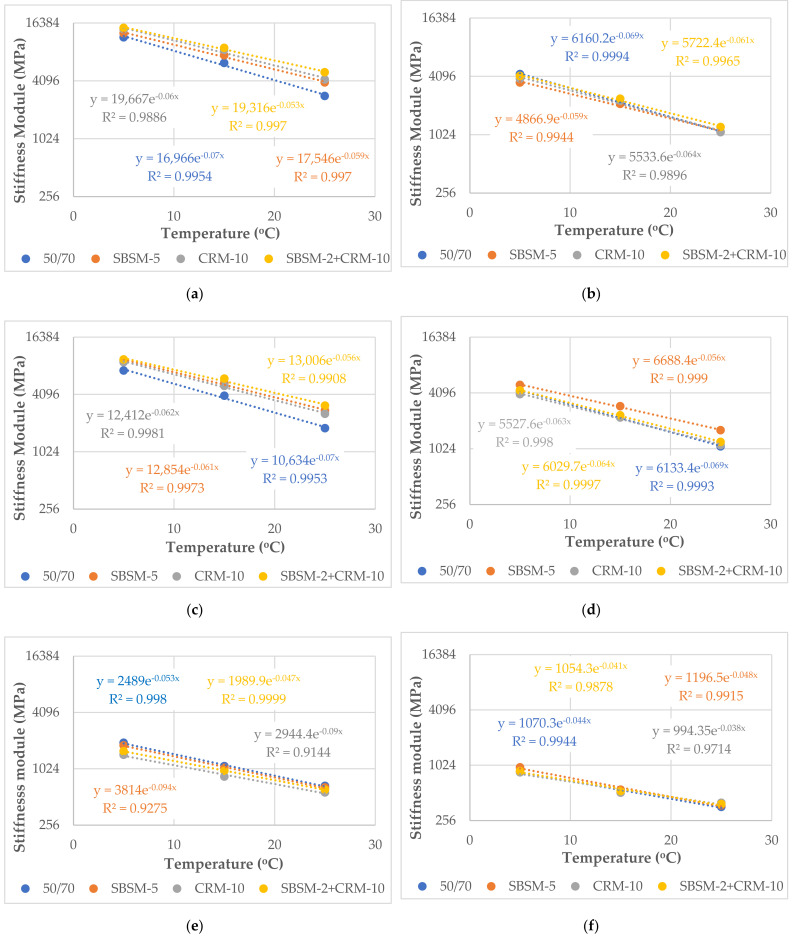
Change of 4PB-PR stiffness modulus values as a function of temperature at 10 Hz for mixtures: (**a**) SMA8, (**b**) PA8, (**c**) SMA8 LA, (**d**) SMA8 LA (10% RG), (**e**) SMA8 LA (20% RG), (**f**) SMA8 LA (30% RG).

**Figure 4 materials-15-05476-f004:**
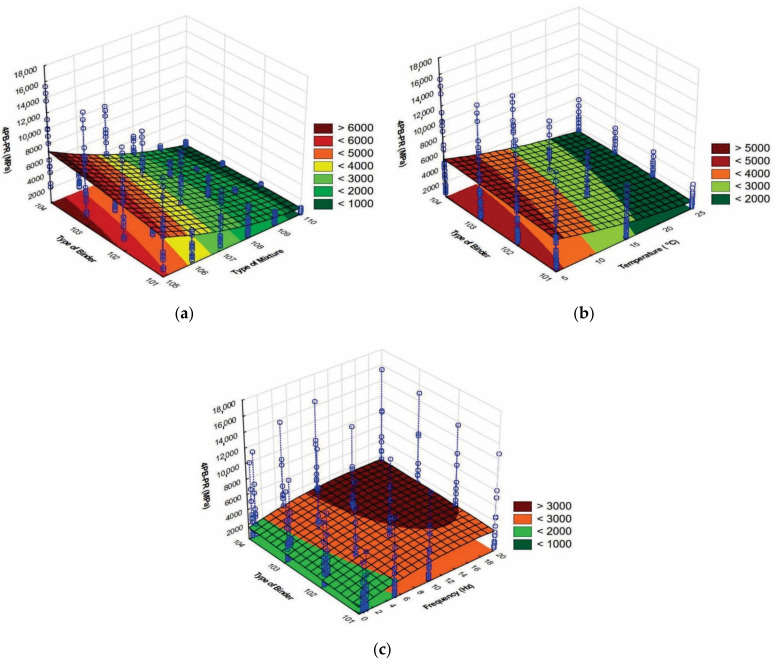
The dependence of the stiffness modulus 4 PB-PR: (**a**) as a function of the type of mixture and binder, (**b**) as a function of the type of binder and temperature, (**c**) as a function of the type of binder and frequency.

**Figure 5 materials-15-05476-f005:**
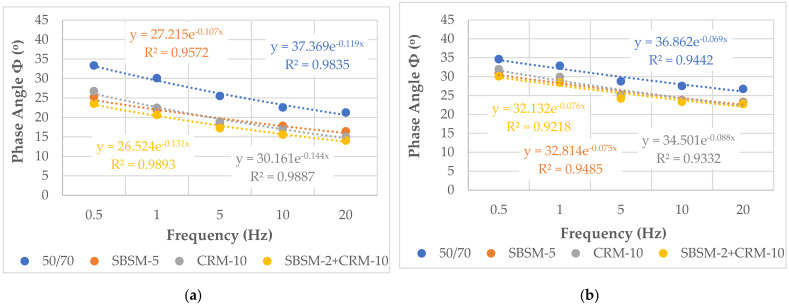
Phase angle, Φ, change at 15 °C at variable frequencies for the mixtures: (**a**) SMA8, (**b**) PA8, (**c**) SMA8 LA, (**d**) SMA8 LA (10% RG), (**e**) SMA8 LA (20% RG), (**f**) SMA8 LA (30% RG).

**Figure 6 materials-15-05476-f006:**
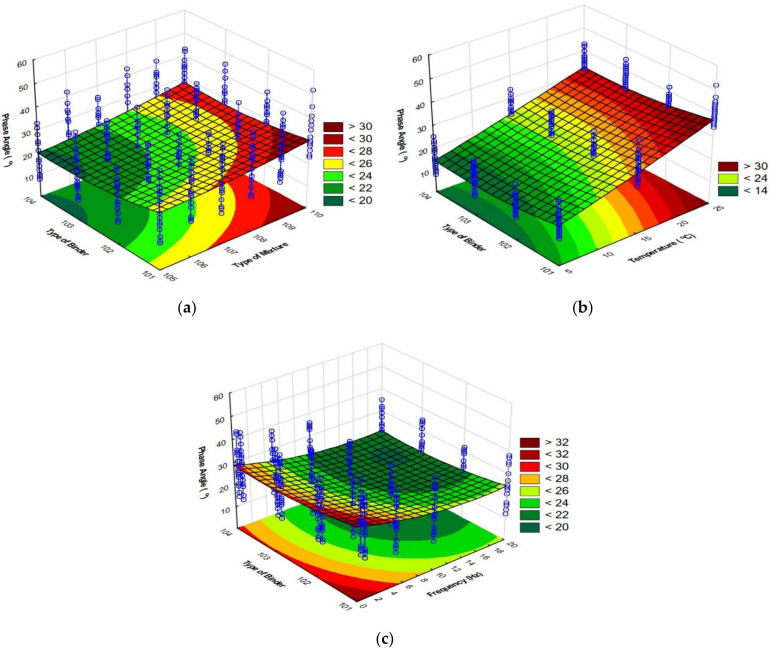
Dependence of the phase angle: (**a**) as a function of the type of mixture and binder, (**b**) as a function of the type of binder and temperature, (**c**) as a function of the type of binder and frequency.

**Table 1 materials-15-05476-t001:** Technical properties of modified binders.

Indexes	Units of Measurement	Type of Binder			
		50/70	SBSM-5	CRM-10	SBSM-2 + CRM-10
		Before RTFOT/ After RTFOT	Before RTFOT/ After RTFOT	Before RTFOT/ After RTFOT	Before RTFOT/ After RTFOT
Penetration:	0.1 mm				
5 °C		11.4/8.4	8.7/6.1	8.7/6.8	7.7/5.6
15 °C		32.8/20.3	20.6/16.2	19.5/14.1	16.2/13.1
25 °C		58.3/44.3	40.2/30.1	40.0/27.8	30.6/24.8
Softening Point	°C	50.8/56.3	78.6/77.8	60.6/68.2	70.7/77.8
Fraass Breaking Point	°C	−14.7/−12.9	−19.3/−17.3	−16.1/−15.5	−17.9/−16.5
Dynamic Viscosity:	Pa·s				
90 °C		11.3/19.3	224.4/258.7	83.6/265.4	292.2/574.1
110 °C		2.2/3.6	25.1/27.7	13.4/39.3	43.7/74.9
135 °C		0.5/0.7	2.5/3.6	2.1/4.8	5.4/8.8

**Table 2 materials-15-05476-t002:** Chemical properties of the SBS triblock polymer.

Chemical Properties	Value
Styrene content (% m)	30
Molecular weight (kg/mol)	151
Bulk density (kg/dm^3^)	0.4
Specific gravity	0.94

**Table 3 materials-15-05476-t003:** Grain size distribution of CR 0/0.8 mm.

Sieve Size (mm)	Percent Passing (%)
1	100
0.71	88.5
0.60	73.5
0.50	38.3
0.43	11.5
0.25	2.3
0.13	0

**Table 4 materials-15-05476-t004:** Sieve analysis of RG 1/4 mm.

Sieve Size (mm)	Percent Passing (%)
4	99.0
3	86.0
2	53.0
1	30.0
0.7	3.0
0.5	0

**Table 5 materials-15-05476-t005:** Grain size distribution (percentage of material passing through the sieve), binder content, air void content, and density of mixtures.

Sieve (mm)	Type of Mixture					
	PA8	SMA8	SMA8 LA	SMA8 LA (10% RG)	SMA8 LA (20% RG)	SMA8 LA (30% RG)
11.2	100.0	100.0	100.0	100.0	100.0	100.0
8	91.2	94.6	92.4	92.8	93.4	93.9
5.6	16.2	48	27.9	31.2	37.2	42.0
4	9.3	38.5	20.5	25.3	31.4	36.4
2	7.7	25.5	15.6	18.6	19.3	19.9
1	7.1	20.2	13.0	14.8	14.6	14.5
0.5	6.6	17.0	11.2	12.2	12.1	12.0
0.25	6.2	14.5	9.9	10.2	10.1	10.0
0.125	5.9	12.7	8.8	8.8	8.7	8.6
0.063	3.9	9.5	6.3	6.3	6.3	6.2
˂0.063	3.9	9.5	6.3	6.3	6.3	6.2
Air voids (%)	23.8	2.89	10.56	11.64	11.99	15.0
Binder content (%)	6.3	6.8	6.8	8.0	10.0	12.0
Maximum density (kg/m^3^)	2.574	2.449	2.528	2.314	2.067	1.949
Bulk density (kg/m^3^)	1.954	2.378	2.261	2.045	1.819	1.651

**Table 6 materials-15-05476-t006:** Results of tests of stiffness modulus 4PB-PR for conventional asphalt mixtures.

Type of Mixture	Type of Binder	Temperature (°C)	Frequency (Hz)				
			0.5	1	5	10	20
SMA8	50/70	5	6509	7666	10,387	11,598	12,743
		15	2685	3369	5332	6279	7091
		25	1019	1299	2322	2849	3352
	SBSM-5	5	8384	9378	11790	12,810	13,557
		15	3844	4546	6576	7532	8411
		25	1636	2039	3284	3936	4605
	CRM-10	5	9217	10,482	12,955	14,033	14,978
		15	4464	5287	7597	8621	9422
		25	1585	2024	3436	4227	4826
	SBSM-2 + CRM-10	5	10,012	11,195	13,580	14,553	15,423
		15	4935	5821	8058	9007	9878
		25	2155	2656	4188	5025	5824
PA8	50/70	5	2372	2798	3872	4325	4587
		15	1037	1274	1923	2247	2542
		25	533	603	915	1096	1155
	SBSM-5	5	2124	2444	3232	3535	3737
		15	1136	1284	1860	2128	2271
		25	540	664	962	1094	1214
	CRM-10	5	2297	2654	3541	3875	4081
		15	1140	1370	2009	2302	2450
		25	570	653	988	1086	1142
	SBSM-2 + CRM-10	5	2529	2926	3577	4135	4521
		15	1255	1486	2145	2407	2521
		25	584	702	1044	1230	1358
SMA8 LA	50/70	5	3963	4709	6550	7286	8055
		15	1616	2025	3281	3961	4478
		25	668	821	1415	1806	2113
	SBSM-5	5	5862	6618	8482	9309	9916
		15	2673	3199	4642	5371	6145
		25	1181	1434	2310	2764	3016
	CRM-10	5	5472	6253	8128	8931	9680
		15	2385	2911	4366	5028	5746
		25	1001	1254	2088	2560	3059
	SBSM-2 + CRM-10	5	6481	7004	8794	9531	10,057
		15	3141	3699	5238	5993	6547
		25	1323	1665	2593	3118	3437

**Table 7 materials-15-05476-t007:** Results of tests of stiffness modulus 4PB-PR for asphalt mixtures with the addition of RG.

Type of Mixture	Type of Binder	Temperature (°C)	Frequency (Hz)				
			0.5	1	5	10	20
SMA8 LA (10% RG)	50/70	5	2432	2908	3920	4298	4491
		15	1052	1272	1908	2233	2539
		25	487	571	917	1084	1318
	SBSM-5	5	3226	3631	4596	4992	5311
		15	1606	1869	2604	2938	3267
		25	782	934	1402	1619	1644
	CRM-10	5	2354	2771	3638	3962	4157
		15	1042	1285	1899	2220	2538
		25	512	629	950	1121	1301
	SBSM-2 + CRM-10	5	2715	3098	3939	4349	4627
		15	1233	1477	2106	2346	2605
		25	567	691	1026	1211	1446
SMA8 LA (20% RG)	50/70	5	1199	1376	1783	1939	2069
		15	621	722	994	1091	1233
		25	424	487	630	672	724
	SBSM-5	5	1188	1341	1692	1819	1970
		15	708	762	985	1046	1110
		25	393	423	595	644	725
	CRM-10	5	943	1082	1398	1442	1638
		15	512	567	778	843	945
		25	306	359	481	571	652
	SBSM-2 + CRM-10	5	1010	1191	1467	1574	1823
		15	569	652	891	973	1044
		25	379	423	537	611	712
SMA8 LA (30% RG)	50/70	5	544	620	800	875	980
		15	324	363	477	527	573
		25	251	277	328	360	417
	SBSM-5	5	596	679	853	971	1097
		15	357	397	505	554	616
		25	258	287	346	375	419
	CRM-10	5	556	614	742	857	951
		15	351	378	479	514	579
		25	274	300	363	398	461
	SBSM-2 + CRM-10	5	543	619	765	883	964
		15	331	362	474	535	586
		25	261	291	357	387	429

**Table 8 materials-15-05476-t008:** Assessment of the significance of influence of temperature, type of mixture, type of modifier and frequency on changes in the stiffness modulus using the ANOVA test.

Effect	Variable: 4PB-PR (MPa); R^2^ = 0.7665; R^2^ adj = 0.7571; Error MS = 2,313,330			
	SS	MS	F	*p*
(1) Type of Mixture (L)	1.041 × 10^9^	1.041 × 10^9^	450.204	˂0.05
Type of Mixture (Q)	4.099 × 10^7^	4.099 × 10^7^	17.719	˂0.05
(2) Type of Binder (L)	1.850 × 10^7^	1.850 × 10^7^	7.998	˂0.05
Type of Binder (Q)	9.433 × 10^5^	9.433 × 10^5^	0.408	0.524
(3) Temperature (L)	6.266 × 10^8^	6.266 × 10^8^	270.847	˂0.05
Temperature (Q)	1.375 × 10^7^	1.375 × 10^7^	5.944	˂0.05
(4) Frequency (L)	1.130 × 10^8^	1.130 × 10^8^	48.859	˂0.05
Frequency (Q)	2.775 × 10^7^	2.775 × 10^7^	11.997	˂0.05
1L·2L	2.636 × 10^7^	2.636 × 10^7^	11.393	˂0.05
1L·3L	3.204 × 10^8^	3.204 × 10^8^	138.517	˂0.05
1L·4L	6.271 × 10^7^	6.271 × 10^7^	27.108	˂0.05
2L·3L	9.426 × 10^5^	9.426 × 10^5^	0.407	0.524
2L·4L	2.192 × 10^4^	2.192 × 10^4^	0.009	0.923
3L·4L	1.100 × 10^7^	1.100 × 10^7^	4.753	˂0.05
Error	7.981 × 10^8^	2.313 × 10^6^		
Total SS	3.419 × 10^9^			

Where: Q—quadratic; L—linear.

**Table 9 materials-15-05476-t009:** Parameters of model describing the dependence of the stiffness modulus on temperature, frequency, type of mixture, and modifier.

Effect	Variable: 4PB-PR (MPa); R^2^ = 0.7665; R^2^ adj = 0.7571; Error MS = 2,313,330					
	Regression Coefficients	Std. Error	t(345)	*p*-Value	−95% Conf. Lmt	+95% Conf. Lmt
Intercept	−361,542	1,030,245	−0.351	0.726	−2,387,895	1,664,810
(1) Type of Mixture (L)	−16,414	8141	−2.016	˂0.05	−32,425	−403
Type of Mixture (Q)	135	32	4.209	˂0.05	72	198
(2) Type of Binder (L)	26,015	17,042	1.526	0.128	−7505	59,535
Type of Binder (Q)	−51	80	−0.639	0.524	−209	106
(3) Temperature (L)	−6973	1093	−6.379	˂0.05	−9123	−4823
Temperature (Q)	4	2	2.438	˂0.05	1	7
(4) Frequency (L)	3822	1238	3.088	˂0.05	1387	6256
Frequency (Q)	−7	2	−3.464	˂0.05	−11	−3
1L·2L	−142	42	−3.375	˂0.05	−224	−59
1L·3L	68	6	11.769	˂0.05	56	79
1L·4L	−34	7	−5.207	˂0.05	−47	−21
2L·3L	−6	9	−0.638	0.524	−23	12
2L·4L	1	10	0.097	0.923	−19	21
3L·4L	−3	1	−2.180	˂0.05	−6	−0

Where: Q—quadratic; L—linear.

**Table 10 materials-15-05476-t010:** Change of the phase angle Φ at variable temperature and frequency for conventional asphalt mixtures.

Type of Mixture	Type of Binder	Temperature (°C)	Frequency (Hz)				
			0.5	1	5	10	20
SMA8	50/70	5	20	18	15	13	12
		15	33	30	25	23	21
		25	43	40	36	36	34
	SBSM-5	5	16	14	11	10	10
		15	25	22	19	18	16
		25	34	33	29	27	26
SMA8	CRM-10	5	16	14	11	10	9
		15	27	22	19	17	15
		25	39	37	32	30	28
	SBSM-2 + CRM-10	5	14	12	10	9	8
		15	24	21	17	16	14
		25	33	31	28	25	24
PA8	50/70	5	24	22	18	17	16
		15	35	33	29	28	27
		25	43	41	39	37	34
	SBSM-5	5	21	20	17	16	15
		15	31	29	25	24	23
		25	39	38	35	33	30
	CRM-10	5	22	20	17	16	14
		15	32	30	25	24	23
		25	39	38	36	34	33
	SBSM-2 + CRM-10	5	20	19	16	15	14
		15	30	29	24	23	23
		25	42	37	34	33	29
SMA8 LA	50/70	5	23	20	17	16	14
		15	35	33	27	25	24
		25	44	42	39	38	37
	SBSM-5	5	17	15	13	12	11
		15	28	25	21	20	19
		25	38	35	32	30	29
	CRM-10	5	19	16	13	13	11
		15	31	28	23	21	20
		25	40	40	34	32	31
	SBSM-2 + CRM-10	5	16	14	12	11	10
		15	26	23	20	18	17
		25	35	34	30	28	27

**Table 11 materials-15-05476-t011:** Change of the phase angle Φ at variable temperature and frequency for asphalt mixtures with the addition of RG.

Type of Mixture	Type of Binder	Temperature (°C)	Frequency (Hz)				
			0.5	1	5	10	20
SMA8 LA (10% RG)	50/70	5	23	20	17	17	16
		15	33	32	28	25	25
		25	44	42	38	37	34
	SBSM-5	5	17	15	13	12	11
		15	27	25	21	18	17
		25	34	33	29	26	24
	CRM-10	5	21	20	17	16	15
		15	32	31	27	24	24
		25	43	40	37	36	34
	SBSM-2 + CRM-10	5	19	18	15	14	13
		15	28	28	23	21	20
		25	43	40	36	34	31
SMA8 LA (20% RG)	50/70	5	22	21	20	19	17
		15	34	32	29	28	27
		25	47	44	42	41	38
	SBSM-5	5	20	19	16	15	15
		15	31	28	26	23	20
		25	42	37	34	33	31
	CRM-10	5	24	21	18	16	13
		15	34	29	28	27	25
		25	41	39	38	37	35
	SBSM-2 + CRM-10	5	23	19	18	16	12
		15	30	28	25	24	21
		25	42	37	36	35	32
SMA8 LA (30% RG)	50/70	5	29	28	26	24	23
		15	41	37	35	32	30
		25	52	45	39	37	35
	SBSM-5	5	25	21	20	16	15
		15	35	33	29	28	24
		25	42	42	38	36	32
	CRM-10	5	25	23	21	19	19
		15	33	32	29	28	22
		25	45	42	39	35	35
	SBSM-2 + CRM-10	5	25	25	22	20	20
		15	34	32	30	28	24
		25	43	42	38	36	35

**Table 12 materials-15-05476-t012:** Assessment of the significance of the influence of temperature, frequency, type of mixture, and binder on changes in the phase angle using ANOVA analysis.

Effect	Variable: Phase Angle Φ (°); R^2^ = 0.9233; R^2^ adj = 0.9202; Error MS = 6.824			
	SS	MS	F	*p*
(1) Type of Mixture (L)	1863.588	1863.588	273.099	˂0.05
Type of Mixture (Q)	40.357	40.357	5.914	˂0.05
(2) Type of Binder (L)	712.693	712.693	104.441	˂0.05
Type of Binder (Q)	288.011	288.011	42.206	˂0.05
(3) Temperature (L)	20,944.945	20,944.945	3069.368	˂0.05
Temperature (Q)	17.735	17.735	2.599	0.108
(4) Frequency (L)	2190.877	2190.877	321.061	˂0.05
Frequency (Q)	513.714	513.714	75.282	˂0.05
1L·2L	53.505	53.505	7.841	˂0.05
1L·3L	10.937	10.937	1.603	0.206
1L·4L	2.502	2.502	0.367	0.545
2L·3L	12.607	12.607	1.848	0.175
2L·4L	0.261	0.261	0.038	0.845
3L·4L	44.807	44.807	6.566	˂0.05
Error	2354.233	6.824		
Total SS	30,701.556			

Where: Q—quadratic; L—linear.

**Table 13 materials-15-05476-t013:** Parameters of the applied model of the dependence of the phase angle, Φ, on temperature, frequency, type of mixture, and binder.

Effect	Variable: Phase Angle Φ (°); R^2^ = 0.9233; R^2^ adj = 0.9202; Error MS = 6.824					
	Regression Coefficients	Std. Error	t (345)	*p*-Value	−95% Conf. Lmt	+95% Conf. Lmt
Intercept	13,119.572	1769.445	7.415	˂0.05	9639.314	16,599.829
(1) Type of Mixture (L)	−47.956	13.981	−3.430	˂0.05	−75.455	−20.457
Type of Mixture (Q)	0.134	0.055	2.432	˂0.05	0.026	0.243
(2) Type of Binder (L)	−206.074	29.270	−7.040	˂0.05	−263.645	−148.503
Type of Binder (Q)	0.894	0.138	6.497	˂0.05	0.624	1.165
(3) Temperature (L)	4.300	1.877	2.290	˂0.05	0.607	7.992
Temperature (Q)	0.005	0.003	1.612	0.108	−0.001	0.010
(4) Frequency (L)	−0.536	2.126	−0.252	0.801	−4.717	3.645
Frequency (Q)	0.030	0.003	8.677	˂0.05	0.023	0.037
1L·2L	0.202	0.072	2.800	˂0.05	0.060	0.344
1L·3L	−0.012	0.010	−1.266	0.206	−0.032	0.007
1L·4L	−0.007	0.011	−0.605	0.545	−0.029	0.015
2L·3L	−0.020	0.015	−1.359	0.175	−0.050	0.009
2L·4L	0.003	0.017	0.195	0.845	−0.030	0.037
3L·4L	−0.006	0.002	−2.562	˂0.05	−0.011	−0.001

Where: Q—quadratic; L—linear.

## Data Availability

Data available in a publicly accessible repository.

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
