# Peer review of "Analysis of the Behavior of Low-Noise Asphalt Mixtures with Modified Binders under Sinusoidal Loading"

_materials, 2022, doi:10.3390/ma15165476_

Round 1

Reviewer 1 Report

This paper reports experimental and statistical analysis results of low-noise asphalt mixtures with modified binders and rubber granulate replacement of aggregate, with respect to stiffness modulus using 4PB-PR testing method. This paper was well written with good flow and logic. It is meaningful to perform a statistical analysis, however, it would be helpful trying to explain the statistical results through fundamental mechanism of binder additives and mixture designs. Also, it is recommend to explain the change of the stiffness modulus on the field performance, especial the aging resistance, so a better practice can be followed.

Author Response

The authors of the paper wish to thank you for valuable comments that helped to improve the quality of this paper.

Point 1:  It is meaningful to perform a statistical analysis, however, it would be helpful trying to explain the statistical results through fundamental mechanism of binder additives and mixture designs.

Response 1The SBS copolymer consists of three-dimensional styrene-butadiene-styrene chains, in which the rigid styrene domains are dispersed in a flexible butadiene matrix that serves as a continuous phase. When SBS is added to bitumen, the styrene blocks, through physical connections, can form three-dimensional networks that enable elastomers to give binders higher stiffness. When crumb rubber is added to the binder, it causes physical swelling and chemical degradation (devulcanization and depolymerization). Maltenes (light fractions of bitumen) are absorbed by rubber network. This process is the main phenomenon that causes the swelling of crumb rubber (three-five times its original volume). In addition, a "gel" structure is formed at the bitumen/rubber interface. At this stage, effective stiffening of the binder takes place, which consequently affects the value of stiffness modulus of asphalt mixtures (Lines 237-248).

Point 2: It is recommend to explain the change of the stiffness modulus on the field performance, especial the aging resistance, so a better practice can be followed.

Response 2: Analysis of the stiffness modulus of asphalt mixtures is important in terms of aging and changes as a function of time. However, in the article, the authors wanted to focus more on the value of stiffness modulus in terms of the type of modifier and the amount of added rubber granulate. The higher the stiffness modulus of each layer, the higher the durability of the entire road pavement structure according to SHRP nomenclature. On the other hand, according to the authors' experience, the change in stiffness modulus as a function of temperature is very important. At lower temperatures, this modulus is recommended to be relatively low to reduce thermal stresses in the mixtures. Conversely, at high temperatures, it is recommended that the modulus be as high as possible. This guarantees that permanent deformations will be minimized. The authors included this in lines 249-256.

In addition, based on the results obtained and the pros and cons, practical recommendations were formulated for the use of the tested mixtures (Point 7 in Conclusions).

Reviewer 2 Report

The ABSTRACT section is well–structured, is informative, can stand alone and covers the content.

The AIMS AND OBJECTIVES  of the research (tests of the stiffness modulus, replacing the mineral aggregate with rubber granulate) are well defined. The purpose of this study was to determine the stiffness modulus of low-noise asphalt mixtures

The TOPICS (low-noise asphalt mixtures with the addition of rubber granulate, type of bitumen modifiers: styrene-butadiene-styrene copolymer and crumb rubber, replacing the mineral aggregate) is an important subjects.

The paper is structured properly and have the basic structure of a typical research paper (INTRODUCTION, MATERIAL , METHODS, RESULTS & DISCUSSIONS, CONCLUSIONS, REFERENCES, etc.). The paper is well–structured and its parts are logically interconnected. Overall, this manuscript is well–written and interesting to read.

The INTRODUCTION section provide the necessary background information needed to understand the paper. The authors give informations on the studied problem.

The MATERIALS section is relatively well described and include detailed information regarding the bitumen binders, the used additives for binders and asphalt mixtures and the low-noise asphalt mixtures (binder content, air void content, grain size composition and density of the designed asphalt mixtures).

The EXPERIMENTAL METHODOLOGY section include all the technical details of the experimental setup, measurement procedure (tests in a wide temperature range: 5C, 15C, 25C, at the following frequencies: 0.5 Hz, 1.0 Hz, 5.0 Hz, 10 Hz, 20 Hz.), and details of how the methods were validated, having in view that due to the fact that asphalt mixture is a typical visco-elastic material, its mechanical properties largely depend on the test temperature and frequency. Overall, this section is technically and fairly detailed.

The body of paper describe the important RESULTS of the research (on testing the stiffness modulus at different frequencies at temperatures, the statistical analysis & regression model, graphical interpretation), followed by several DISCUSSIONS, based on the analysis of the parameters. The statistical analysis of the obtained results was started with the significance test 284 using the ANOVA analysis of variance (STATISTICA software). The authors do a very good job of presenting their results and demonstrate the suitability of the method.

The CONCLUSION section succinctly summarize the major points of the paper, derived from the RESULTS and the DISCUSSIONS. The authors fairly concludes in just a few sentences given the rich discussion in the body of the paper.

The list of REFERENCES is long and relatively well chosen. The entire BIBLIOGRAPHY is current (the oldest being from 2007).

All the 11 TABLES are representative and all the 2 FIGURES & 4 series of GRAPHS have good qualities.

Author Response

The authors of the paper wish to thank you for valuable comments.

Reviewer 3 Report

The manuscript presents a study on the behaviour of low-noise asphalt mixtures with two modified binders under sinusoidal loading, based on the results of four point bending tests of prismatic samples made of several compositions, and a posterior data analysis. The topic and problem under study are well introduced and supported by suitable references. The study of different modified bitumen binders and asphalt mixtures is an addition to the research field, and the results seem relevant, even though the innovative character of the work could be further emphasized. The research methodology is adequate, but the adopted methods need to be better explained, particularly the framework of the analytical tests. The results sound accurate, are relatively well presented and discussed, despite not being so concisely reported, and support the conclusions. The manuscript is in general well structured and written, but there is room for improvement. A major comment about the manuscript is that it presents many results, but no significant recommendations for application practice are given. Beyond the general comments here, specific comments are provided below.

1. General: Acronyms RG, SBS and CR are not defined in the main text. Consider adding an acronyms list. The paper is not free of errors and careful proofreading is required.

2. Intro, line 90: Please clarify the meaning of the expression ‘durability of low temperature cracking’.

3. Add an introductory note in Section 2, before 2.1, referring to what will be presented in the section.

4. The results in Table 1 should be discussed a little in the text. Same for Table 5 and Figure 1. For example, can the Authors find a pattern in Figure 1 for the particle size distribution related to certain mixtures?

5. Section 3 only has Subsection 3.1, so a suggestion is to add an introductory subsection, i.e., 3.1 Intro and then 3.2. This is a rather short section when compared to the others. The test layout can be presented in more detail.

6. Add and introductory note in Section 4, before 4.1, referring to what will be presented in the section.

7. Table 6 has many rows as it spans 3 pages. A suggestion is to divide it into several tables, e.g., for each pair of Types of Mixtures (SMA8 and PA8, SMA8 LA and SMA8 LA (10% RG), etc.).

8. Line 240: The cause-effect is not so clear in following sentence: ‘the stiffness modulus begins to increase rapidly with increasing thermal stresses, which contributes to the formation of low-temperature cracking’. Please improve.

9. Line 253: ‘It is worth emphasizing that the addition of another 10% of RG to the SMA8 LA mixture causes a decrease in the modulus value by approximately 50% practically in the entire range of analyzed temperatures’. It is not so clear which specific mixture this phrase refers to.

10. Section 4.1: The framework of the ANOVA tests whose results are presented in Tables 7 and 8 should be further explained. Is there any theory for fitting an exponential relationship to the graphs in Figure 3? Please give an explanation.

11. Section 4.2 is similar in structure and content to Section 4.1, therefore, the same notes as above apply. In these two sections, a more concise report of the results would be beneficial.

12. Conclusions: The statements in points 1 and 2 are partially overlapped. Please improve. Based on the results obtained and the pros and cons, practical recommendations should be made regarding the application of the tested mixtures.

Author Response

The authors of the paper wish to thank you for valuable comments that helped to improve the quality of this paper.

Point 1: General: Acronyms RG, SBS and CR are not defined in the main text. Consider adding an acronyms list. The paper is not free of errors and careful proofreading is required.

Response 1: All acronyms were defined before they appeared in main text (line 51, line 101).

Point 2:  Intro, line 90: Please clarify the meaning of the expression ‘durability of low temperature cracking’.

Response 2: Low values of the stiffness modulus at low test frequencies testify to poor pavement resistance to permanent deformation, while its high values at higher frequencies negatively affect the low-temperature performance at winter conditions. 

Point 3: Add an introductory note in Section 2, before 2.1, referring to what will be presented in the section

Response 3: Articles in Materials use this form of output presentation and, according to the authors, no additional notes are needed.

Point 4: The results in Table 1 should be discussed a little in the text. Same for Table 5 and Figure 1. For example, can the Authors find a pattern in Figure 1 for the particle size distribution related to certain mixtures?

Response 4: The results in Table 1, Table 5 and Figure 1 are presented in accordance with the principle used to describe materials and asphalt mixtures.

Point 5: Section 3 only has Subsection 3.1, so a suggestion is to add an introductory subsection, i.e., 3.1 Intro and then 3.2. This is a rather short section when compared to the others. The test layout can be presented in more detail.

Response 5: Sections 2 and 3 have been combined. The title of the new section is "Materials and Methods".

Point 6: Add and introductory note in Section 4, before 4.1, referring to what will be presented in the section

Response 6: According to the authors, the title of Section 4 clearly describes what will be presented in this Section.

Point 7: Table 6 has many rows as it spans 3 pages. A suggestion is to divide it into several tables, e.g., for each pair of Types of Mixtures (SMA8 and PA8, SMA8 LA and SMA8 LA (10% RG), etc.).

Response 7: Table 6 has been divided into two tables: one for conventional asphalt mixtures (SMA8, SMA8 LA, PA8), the other for asphalt mixtures with the addition of rubber granulate (SMA8 LA [10% RG], SMA8 LA [20% RG], SMA8 LA [30% RG]).

Point 8: Line 240: The cause-effect is not so clear in following sentence: ‘the stiffness modulus begins to increase rapidly with increasing thermal stresses, which contributes to the formation of low-temperature cracking’. Please improve

Response 8: Inverse changes are observed when the test temperature is decreased: the value of stiffness modulus begins to increase rapidly. This indicates that there is an increase in thermal stresses in the visco-elastic material. When the thermal stress value exceeds the tensile strength of the material, the initiation of low-temperature cracking occurs.

Point 9: Line 253: ‘It is worth emphasizing that the addition of another 10% of RG to the SMA8 LA mixture causes a decrease in the modulus value by approximately 50% practically in the entire range of analyzed temperatures’. It is not so clear which specific mixture this phrase refers to.

Response 9: The analysis concerns the change in the parameters of the SMA8 LA as a reference mixture. Authors analyzed the effect of the addition of rubber granulate on the change of the stiffness modulus of this mixture as a function of temperature.

Point 10: Section 4.1: The framework of the ANOVA tests whose results are presented in Tables 7 and 8 should be further explained. Is there any theory for fitting an exponential relationship to the graphs in Figure 3? Please give an explanation.

Response 10: The obtained regression model (Equation 10) allows to predict with good precision (about 76%) the stiffness modulus of the analyzed mixtures with SBS and crumb rubber modified binder (Figure 3). The values of stiffness modulus are essential to mechanistic models for fatigue life design of road pavements.

Point 11: Section 4.2 is similar in structure and content to Section 4.1, therefore, the same notes as above apply. In these two sections, a more concise report of the results would be beneficial.

Response 11: The obtained regression model (Equation 12) allows to predict with very good precision (about 92%) the stiffness modulus of the analyzed mixtures with SBS and crumb rubber modified binder (Figure 5).

Point 12: Conclusions: The statements in points 1 and 2 are partially overlapped. Please improve. Based on the results obtained and the pros and cons, practical recommendations should be made regarding the application of the tested mixtures.

Response 12: The authors propose to leave points 1 and 2 in conclusions, because point 1 is about the effect of the type of modifier for asphalt binder on the parameters of low-noise asphalt mixtures and point 2 is about the effect of the content of rubber granulate on the change in stiffness modulus of low-noise asphalt mixtures. Based on the results, authors formulated practical recommendations for the use of the tested mixtures (Point 7).

Round 2

Reviewer 3 Report

The manuscript presents a study on the behaviour of low-noise asphalt mixtures, including experimental and analytical investigations. It has been resubmitted following a previous review. The Authors have in general considered all the raised comments and the paper is improved. This reviewer has just a few comments to be considered.

1. A brief introductory note to Sections 2 and 3 can be made to improve readability, or else the content of each section may be briefly mentioned at the end of the general Intro.

2. Section 2.3: The particle size distribution of the tested mixtures is well illustrated, but its influence on the mixture behaviour is not so well discussed. Please give a short note.

3. Mention should be made in the text to the exponential fit of the plots in Figures 3 and 5.

Author Response

The authors of the paper thank you for your valuable comments.

Point 1: A brief introductory note to Sections 2 and 3 can be made to improve readability, or else the content of each section may be briefly mentioned at the end of the general Intro.

Response 1: The authors added a brief explanation at the end of the general Intro (Lines 109-122).

Point 2: Section 2.3: The particle size distribution of the tested mixtures is well illustrated, but its influence on the mixture behaviour is not so well discussed. Please give a short note.

Response 2: The authors provided a short note after Figure 1 (Lines 181-187).

Point 3: Mention should be made in the text to the exponential fit of the plots in Figures 3 and 5

Response 3: In the text, the exponential fit of the plots in Figures 3 and 5 was mentioned (Lines 341-343 and 445-447).